# Prefrontal Cortex Hemodynamics and Functional Connectivity Changes during Performance Working Memory Tasks in Older Adults with Sleep Disorders

**DOI:** 10.3390/brainsci13030497

**Published:** 2023-03-15

**Authors:** Jiahui Gao, Lin Zhang, Jingfang Zhu, Zhenxing Guo, Miaoran Lin, Linxin Bai, Peiyun Zheng, Weilin Liu, Jia Huang, Zhizhen Liu

**Affiliations:** 1College of Rehabilitation Medicine, Fujian University of Traditional Chinese Medicine, Fuzhou 350122, China; 2National-Local Joint Engineering Research Center of Rehabilitation Medicine Technology, Fujian University of Traditional Chinese Medicine, Fuzhou 350122, China

**Keywords:** working memory, functional near-infrared spectroscopy, sleep disorders, functional connectivity, hemodynamic, prefrontal cortex

## Abstract

Objective: Older adults with sleep disorders (SDs) show impaired working memory abilities, and working memory processes are closely related to the prefrontal cortex (PFC). However, the neural mechanism of working memory impairment in older adults with SD remains unclear. This study aimed to investigate changes in PFC function among older adults with SD when carrying out the N-back task by functional near-infrared spectroscopy (fNIRS). Method: A total of 37 older adults with SDs were enrolled in this study and matched with 37 healthy older adults by gender, age, and years of education. Changes in PFC function were observed by fNIRS when carrying out the N-back task. Results: The accuracy on the 0-back and 2-back tasks in the SD group was significantly lower than that in the healthy controls (HC) group. The oxygenated hemoglobin (oxy-Hb) concentration of channel 8 which located in the dorsolateral prefrontal cortex (DLPFC) was significantly reduced in the SD group during the 2-back task, and the channel-to-channel connectivity between the PFC subregions was significantly decreased. Conclusions: These results suggest that patients with sleep disorders have a weak performance of working memory; indeed, the activation and functional connectivity in the prefrontal subregions were reduced in this study. This may provide new evidence for working memory impairment and brain function changes in elderly SDs.

## 1. Introduction

The prevalence of sleep disorders (SDs) in older adults over the age of 60 has been estimated to be 30–40% [1,2], which leads to a decline in cognitive function later [3,4,5]. Impaired working memory is a common symptom of SDs. Impairment of working memory is mainly manifested as the inability to store, maintain, and manipulate information from the outside in a short period, which negatively impacts the everyday living functions of older adults [6,7,8,9,10]. Thus, exploring the underlying impairment is vital for early diagnosis in older adults with SDs.

The prefrontal cortex (PFC) is a critical brain region that modulates higher brain functions, including working memory, attention, and reasoning. Neuroimaging studies have demonstrated that several brain regions, including the PFC, play a vital role in the process of working memory [11,12,13]. The prefrontal cortex can be subdivided into subregions such as the dorsolateral prefrontal cortex (DLPFC), ventrolateral prefrontal cortex (VLPFC), frontopolar cortex (FPC), and orbitofrontal cortex (OFC). The results of a meta study also showed that activity in various frontal subregions, including the DLPFC, VLPFC, and FPC increased during working memory tasks in healthy individuals [14], and that a higher activation maintained a higher working memory performance in older adults. OFC impairment was also associated with deficits in n-back neuropsychological tests involving working memory, suggesting that prefrontal subregion activity is a relevant neural factor in detecting working memory [15]. Existing research on working memory suggests that connectivity between the PFC and other brain regions is positively correlated with working memory performance [16]. The reduced functional connectivity (FC) of PFC subregions may underlie the deficits in working memory in patients with SDs.

In functional magnetic resonance imaging (fMRI) studies, they have observed that sleep deprivation has resulted in reduced working memory performance and abnormalities in PFC activation, spontaneous activity, and functional connectivity [17]. Primary insomnia alters the resting-state FC between the parietal and frontal lobes associated with spatial working memory [18] and decreases the activation in working memory brain-related areas, and it significantly decreases the regulation of the right DLPFC with increasing task difficulty [19]. Taking the above studies into consideration, it is suggested that sleep problems cause changes in PFC function related to working memory; however, the functional connectivity analyses in these studies were mostly based on MRI resting states. Changes in early neurodegenerative processes are better measured during task performance than that at rest state. Therefore, changes in cranial neural function need to be investigated during task completion to detect early changes in working memory associated with sleep disturbances and to identify the early neurodegenerative processes underlying these changes [20]. There are some studies that have explored sleep-related PFC functional connectivity based on working memory tasks, but most of these studies have focused on middle-aged and young adults. Patients with obstructive sleep apnea (OSA) syndrome had reduced effective connectivity from the right FPC and DLPFC to the left PFC during the verbal version of the n-back task [21]. As older adults have a high incidence of SDs, the neural mechanism of SDs needs to be investigated in elderly individuals. Most studies of working memory in geriatric populations with SDs thus far have focused on behavioral manifestations, and further studies on the activation of PFC subregions and interregional functional connectivity are scarce in this population.

At present, PFC neuroimaging in older adults with SDs has not yet been studied during working memory tasks. Functional near-infrared spectroscopy (fNIRS) is a new brain mechanism functional imaging technology. Compared with other detection tools, fNIRS has the advantages of relatively low environmental requirements, greater tolerance for motion artifacts, no radiation, and enabling participants to complete assigned tasks in a reasonably relaxed state in the real environment [22], making it an elder-friendly neuroimaging technique. Monitoring changes in the Oxygenated hemoglobin concentration in the PFC while completing cognitive tasks could objectively reflect cognitive levels.

In this study, fNIRS was used to detect changes in PFC hemodynamics and FC during the N-back process of working memory in older adults with SDs. We expected that (1) older adults with SDs would show a decline in working memory performance compared to HCs; (2) older adults with SDs would have reduced oxygenated hemoglobin (oxy-Hb) concentrations in the PFC subregion, and weak interrelationships within PFC subregions. To our knowledge, this study is the first to use fNIRS to explore the correlation of PFC subregions with working memory alterations in older adults with SDs, which may contribute to a further understanding of the neural mechanisms underlying working memory deficits in older adults with SDs.

## 2. Materials and Methods

### 2.1. Trial Design and Participants

This study was an exploratory study comparing the development of older adults with or without SDs and investigated the association between PFC hemodynamic responses during working memory tasks in older adults with SDs. We recruited 37 older adults with SDs in the communities of Fuzhou city, China, and matched them with 37 healthy older adults by gender, age, and years of education.

The research was conducted in accordance with the Declaration of Helsinki and was approved by the Ethics Committee of the Affiliated Rehabilitation Hospital of Fujian University of Traditional Chinese Medicine (2022KY-024-01). All participants gave written informed consent after a complete explanation of the procedure.

### 2.2. Diagnostic Criteria

The inclusion criteria for the SD group were as follows: (1) meeting the diagnostic criteria of primary insomnia in the International Classification of an SD, including difficulty falling asleep or difficulty maintaining sleep, adequate opportunity to sleep, impaired daytime functioning, an SD occurring at least three times per week and present for the last 3 months; (2) Pittsburgh Sleep Quality Index (PSQI) score > 5; (3) 65–70 years; (4) Montreal Cognitive Assessment (MoCA) score ≥ 26; and (5) right-handed.

The exclusion criteria for the SD group were as follows: (1) taking medications for sleep disorder (2) patients with depression (Geriatric Depression Scale-15 score ≥ 9); (3) patients with serious stroke complications; and (4) patients with severe heart, liver, kidney failure, malignant tumor, and other significant diseases.

The inclusion criteria for the healthy control (HC) group were as follows: (1) did not meet the diagnostic criteria of primary insomnia; (2) PSQI score ≤ 5; (3) 65–70 years old; (4) MoCA score ≥ 26; and (5) Right-handed.

The exclusion criteria for the HC group were the same as those used for the SD group (see criteria 1–3).

#### 2.2.1. PSQI

The PSQI is a self-administered questionnaire which assesses sleep quality and disturbances. Nineteen individual items generate seven “component” scores: subjective sleep quality, sleep latency, sleep duration, habitual sleep efficiency, sleep disturbances, use of sleeping medication, and daytime dysfunction. A global PSQI score >5 yielded a diagnostic sensitivity of 89.6% and specificity of 86.5% in distinguishing good and poor sleepers [23].

#### 2.2.2. MoCA

The MoCA scale was applied to assess the neuropsychological situation of all participants. We selected the MoCA, which is a one-page, 10-min, 30-point screening test to identify individuals with MCI. It includes the testing of visual space and executive functions, naming, memory, attention, language, abstract thinking, calculation, and orientation. A MoCA score ≥26 indicates normal cognition [24].

### 2.3. N-Back Task Design

The N-back task was presented on a computer using E-Prime3.0, and fNIRS recorded the hemodynamic response changes in the PFC during the N-back task. N-back tasks include 0-back, 1-back, and 2-back tasks. The ratio of a target stimulus to a nontarget stimulus was 1:2, the display time of each stimulus was 1000 ms, and the delay between the display times was 2000 ms. The rest period between each block was 30 s, plus an extra 10 s of instructions before each block. The participants pressed the “1” key on the keyboard for the target stimulus and the “2” key on the keyboard for the nontarget stimulus. Each condition occurred three times, and these blocks appeared in a pseudorandomized order (Figure 1). Participants required practice before any formal measurements to ensure they fully understood the task. The reaction time (RT) and accuracy were recorded during the task.

### 2.4. FNIRS Data Acquisition

Hemodynamic signals were acquired from each dyad simultaneously using a multichannel high-speed, continuous-wave system (LABNIRS, Shimadzu Corp, Kyoto, Japan) consisting of different laser diodes with three wavelengths of 780, 805, and 830 nm. The 57 channels consisted of 18 sources and 18 detectors (Figure 2). According to the international 10–20 system, CH42 were positioned directly at fpz. The coordinates of all probe positions and anatomical landmark positions (Cz, Nz, AL, AR) were obtained using a 3D digitizer, and the coordinate of a fNIRS channel was computed from the positions of the sources and detectors using NIRS-SPM software [25]. The corresponding relationship between brain regions and Brodmann regions was obtained, and the brain regions were divided. A total of 43 channels located in the PFC were divided into four anatomical regions: DLPFC (CH1, CH2, CH3, CH4, CH5, CH7, CH8, CH9, CH10, CH11, CH12, CH13, CH14, CH19, CH23, CH29, CH34, CH50, CH55); VLPFC (CH17, CH18, CH24, CH25, CH28, CH35, CH39, CH45); FPC (CH20, CH21, CH22, CH30, CH31, CH32, CH33, CH40, CH41, CH42, CH43, CH44); and OFC (CH51, CH52, CH53, CH54) (Figure 2).

### 2.5. fNIRS Data Analysis

fNIRS data were processed and analyzed with Homer2(David Boas, Jay Dubb, Ted Huppert, Meryem Ayse Yucel, Boston, MA, USA) using MATLAB. The participants’ raw data were converted to a Homer2-compatible format (*nirs) and then pre-processed. For pre-processing (1) noisy channels were pured by enPruneChannels function [dRange(1) = 1 × 10^−2^, dRange(2) = 3 × 10, and SNRthresh = 2] (Table 1); (2) the raw NIRS light intensity was converted to an optical density (OD) signal using the hmrIntensity2OD function; (3) filtration: the hmrMotionArtifactByChannel function in the Homer2 was used to automatically detect motion artifacts. Then the hmrMotionCorrectSpline function was used to create and subtract the splines as well as to add the vertical shift to realign the data. We then applied a bandpass filter with cutoff frequencies of 0.01–0.5 Hz to remove physiological noise; (4) the filtered optical density data were converted into oxy-Hb, deoxy-Hb, and total-Hb concentrations by applying the modified Beer–Lambert law using the Homer2′ hmrOD2Conc function; and (5) The Homer2 hmrBlockAvg function were used to complete the block average. After preprocessing, the individual hemoglobin concentration for the three tasks was obtained by mean time series blocks. The mean hemoglobin concentration at the group level was then acquired by averaging the hemoglobin concentration changes of all individuals.

fMRI studies mainly identified functional resting-state networks in the 0.01–0.10 Hz frequency range and the fNIRS data for the FC analysis were filtered in this range without step 5 during preprocessing. The various block time processes were connected for each N-back load, and the coherence between all channel pairs was performed with the matlab function code mscohere [26]. Then, the coefficients of all channel pairs were z-transformed, and an independent samples *t*-test was used for between-group comparisons. We only observed oxy-Hb changes during the N-back task because the oxy-Hb signal can better reflect changes in local cerebral blood oxygen and provide a better signal-to-noise ratio than deoxy-Hb [27]. All three NIRS parameters (HbO\HbR\HbT) are presented in the Appendix A.

### 2.6. Sample Sizes

Sample sizes were calculated by G*Power 3.1(Axel Buchner, Edgar Erdfelder, Franz Faul, Albert-Georg Lang, North Rhine-Westphalia, Germany), using a ’means: Difference between two dependent means (matched pairs)’ test. The smallest sample sizes were estimated as 74, with an alpha level of 5% and power of 80%, and an effect size of 0.33 (calculated based on a previous study [28])

### 2.7. Statistical Analysis

The data were analyzed using IBM SPSS Statistics version 24.0. Categorical data were analyzed by the chi-square test. The Shapiro–Wilk test was used to assess the normality of the continuous numerical variables with, and then the independent samples *t*-test was performed for those data that were normally distributed, and the Mann–Whitney test was performed for those data that were not normally distributed. An independent sample *t*-test (or Mann–Whitney test for non-normal data) was performed on the accuracy and RT of the N-back, hemodynamic concentration, and coherence data of both groups. Data are expressed as mean ± standard deviation or median (first quartile, third quartile). False discovery rate (FDR) correction was used to correct fNIRS data (all 43 channels) to control multiple comparisons and avoid the occurrence of a class of errors. Lastly, the effect sizes were calculated using Cohen’s d test [29] or using rank-biserial correlation for the Mann–Whitney U test [30]. *p* ≤ 0.05 considered the difference to be statistically significant.

## 3. Results

### 3.1. Demographics

The demographic statistics are shown in Table 2. The age, years of education, or gender did not significantly differ between the SD group and HC group (*p* > 0.05).

### 3.2. N-Back Performance

The accuracy of the 0-back task was significantly lower in the SD group compared to that in the HC group. (*p* = 0.03). However, during the 1-back test, both groups performed with high accuracy and did not differ significantly (*p* = 0.80). During the 2-back test, the SD group performed no better than the HC group and made a significant difference (*p* = 0.05). During the 0-back, 1-back, and 2-back conditions, the reaction time of the SD group was not significantly different from that of the HC group (Table 3).

### 3.3. The PFC’s Hemodynamic Response during N-Back Task

We observed all 43 channels in the PFC, which were located with a 3D digitizer. During the 2-back test, the oxy-Hb concentration in the CH8 (located in DLPFC) of older adults with SDs was significantly reduced compared to that of the HC group (*p* < 0.01 uncorrected, *p* < 0.01, FDR corrected). Figure 3 shows the difference in the hemodynamic response of CH8 with a significant statistical effect. The hemodynamic alterations did not significantly differ between groups under other load conditions (Appendix A).

### 3.4. Functional Connectivity of Prefrontal Areas

Figure 3 shows the oxy-Hb correlation matrix maps within the HC and SD groups. This finding demonstrated that the FC of channel pairs had a significant coherence in the PFC regions during the 2-back test. FC in channel pairs in DLPFC-FPC [(22,2), (22,9), (22,10), (22,14), (22,29), (42,9), (42,34) (23,30), (23,31), (23,32), (23,33)], VLPFC-FPC [(22,28), (22,35), (42,17), (42,35)], FPC-OFC [(22,51)], DLPFC-VLPFC [(23,17), (23,28), (23,35)], DLPFC-OFC [(23,51), (23,52)], FPC [(22,32), (22,33)], DLPFC [(23,9), (23,10), (23,14), (23,29), (23,34)] was significantly stronger in the HC group (Z = 0.33 to 0.44, *p* < 0.05, FDR corrected) than in the SD group (Z = 0.21 to 0.30) (Figure 4). No pairs in the SD group were found to be significantly stronger than in the HC group. During the 0-back, 1-back conditions, the FC of the SD group was not significantly different from that of the HC group.

## 4. Discussion

The major finding of this study is that compared with the HC group, the performance of 2-back tests related to working memory were decreased in the SD group. Furthermore, the concentration of oxy-Hb in the channel 8 (located in DLPFC) was lower in the SD group than that in the HC group when performing the 2-back task. Moreover, we observed that the PFC subregions’ channel-to-channel connectivity of SDs was weaker than that of HCs. It was suggested that SDs were unable to activate and coordinate the use of multiple PFC subregions to perform the working memory task.

The N-back task has been used in many human studies to investigate the neural basis of the PFC during working memory processes [14,31]. The 2-back accuracy of the SD group decreased significantly in our study, indicating that SDs affect the working memory performance of older adults. This finding was consistent with those of previous reports [9,32,33] showing that short sleep duration or circadian rhythm disruptions may reduce working memory capacity [34]. There was no significant difference in 1-back accuracy, which may be due to the fact that the 2-back task requires a greater brain load than the 1-back task in terms of working memory. As the difficulty of the N-back tasks increases, so does the mental workload, and SD may have a strong effect on the cognitive state. The 0-back task required sustained attention but no working memory load [35]. The significant difference in 0-back between the two groups in our findings may be evidence that sleep disturbance has a negative effect on attention in older adults. For this result, adding an exploration of attention is considered in the follow-up study [36]. This study provides more evidence that SDs negatively impact on working memory performance. However, another study found no significant decline in the N-back performance in older adults with sleep-disordered breathing, which contradicts the results obtained from our research [37]. This difference may be due to differences in the types of subjects between the previous study and our study.

The n-back correct rate in our behavioral results for both SD and HC showed a decreasing trend with increasing working memory load, which is consistent with existing studies [38,39]. There was no significant difference between the 0-back and 1-back reaction time groups, and there was a trend toward a worse reaction time in the SD group compared to the HC group. However, in the more difficult 2-back test of our findings, SD outperformed HC in response time. For this result, some researchers have suggested that subjects make a “trade off” between speed and correctness rate. Accordingly, we suggest that the more difficult 2-back required more cognitive ability, which may have led the HC group to try to maintain correctness and “trade off” between response time and correctness, prolonging response time [40].

Our study compared the hemodynamic concentration changes in the HC and SD groups during the working memory process and found that the oxy-Hb concentration in the CH8 (located in DLPFC) was significantly lower in the SD group during the 2-back task. Available studies have indicated that multiple brain regions are involved in the process of working memory, especially in the PFC [41,42]. During working memory tasks, activated neurons increase the demand for oxygen and then increase the hemodynamic response due to the increased neural electrical response in the prefrontal lobe [43]. SDs will cause functional abnormalities in the brain, by reducing neuronal activation during brain activity, resulting in abnormal brain activity, especially the decreased activation of the PFC subregions, which affects working memory performance [44,45].

The fMRI study by Drummond [19] observed a neural correlation between the working memory performance of patients with primary insomnia and good sleepers. Moreover, they showed that patients with primary insomnia exhibited decreased activation in the frontal regions when performing working memory tasks. Patients with OSA have impaired working memory, and this impairment is associated with a disproportionate impairment of DLPFC function [46]. A fMRI study showed the activation of both the left DLPFC and right VLPFC decreased significantly after 30 h of sleep deprivation in 33 men [47]. In our results, older adults with SDs did not show the same activation of task-related prefrontal regions as HCs. In general, these results suggest that older adults with SDs failed to recruit sufficient DLPFC resources for working memory tasks.

However, the oxy-Hb level did not significantly differ between groups during the 0-back and 1-back conditions in this study, for the reason that task-induced brain activation depends on the ability to perform tasks. Brain area activation is related to the difficulty of the task; the more difficult the task is, the more the degree of task-related activation increases. The 0-back and 1-back are cognitive tasks in low and medium load, which caused a small degree of activation. So, the differences in activation between groups were not statistically significant. A recent systematic review showed that patients with SDs performed worse on attention tasks, which is the same situation as in our study where there was a significant difference in the 0-back accuracy rate [48]. Previous studies have shown that during the performance of a working memory task, participants performing a 1-back rather than a 0-back task had brain activity in working memory areas [49]. Moreover, another study also showed significantly longer feature path lengths in the 2-back task compared to the 0-back task [50]. Combining the above studies, we concluded that the 2-back task was more challenging than the 0-back task, and that the 0-back required less cognitive demand and less PFC activation, so although attention was modulated by PFC and there was a significant difference in 0-back accuracy between the two groups of subjects in this study, the difference in PFC subregions activation was not significant.

The various anatomical areas involved in processing tasks must be able to communicate and synchronize effectively for the system to function properly. FC is a description of active synchronization between remote cortical regions [51]. A working memory study found that young adults who performed better on the task had a higher FC than older adults who performed poorly on the task [52]. As the subprocess of cognitive control, it was proved that working memory depended on FC in the frontal cortex by lots of studies [53,54]. A number of studies have also shown that people with SDs appear differences in FC during resting states and neuropsychological tests of complex information processing [55,56]. The effects of sleep and sleep deprivation on connectivity signs in different brain regions were observed by resting-state electroencephalography (EEG) and sleep deprivation strongly influenced FC in the PFC [57]. These findings agree with those of the present study. We provide new evidence to support that, older adults with SDs exhibit abnormal FC in the process of working memory tasks. In particular, we found a reduction between PFC subregions connectivity in the SD group during the 2-back task.

In addition, the population involved in this study was made up of older adults with SDs who had a MoCA score of at least 26 (relatively normal). The study suggests the presence of overall cognitive impairment including working memory impairment during aging [58,59]. With this influence, we excluded subjects with possible overall cognitive impairment from our study, which allows us to focus our study on the effects of SDs on working memory performance and functional imaging in older adults. Our follow-up study considers further analyses by tasks that can segment different components of working memory, including encoding, storage, and retrieval. Follow-up studies may also be conducted by exploring the relationship between objective sleep recording as well as structure and working memory performance in older adults with SDs.

### Limitation

This study was subject to certain limitations. The PFC is the critical brain area in working memory, and our fNIRS measurements were limited by the number of probes and only covered the PFC, but other regions are also involved in modulating working memory. Thus, future studies should also consider a broader range of brain cortices associated with cognition. The small difference in N-back performance between the two groups is considered to be due to the small sample size, which we have previously estimated by referring to related studies, and therefore we will consider increasing the sample size in future studies to make the results more convincing. In addition, fNIRS can only measure superficial cortex activity but cannot detect subcortical structures that cannot be reached by near-infrared light. Future studies should combine detection with other imaging examinations, such as EEG and MRI, to improve its time resolution and spatial resolution.

## 5. Conclusions

This study is the first to use fNIRS to evaluate the PFC subregions’ function of older adults with SDs during the N-back task. In summary, the reduction in sleep-related activation and FC of the PFC subregions during working memory execution may indicate a specific vulnerability of these regions to SDs, and these results confirm our hypothesis.

## Figures and Tables

**Figure 1 brainsci-13-00497-f001:**
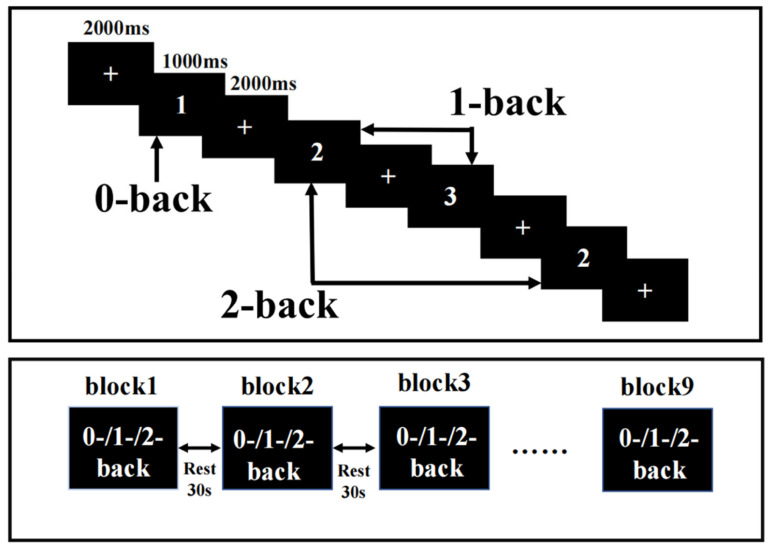
Task design. N-back task with three loads: 0-back, 1-back, and 2-back. A response was required whenever the current stimulus matched the number 1 and the stimulus one or two positions back in the sequence.

**Figure 2 brainsci-13-00497-f002:**
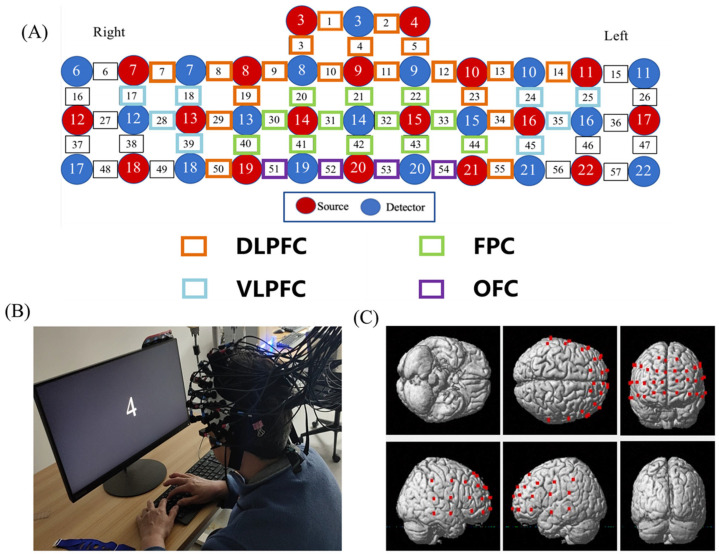
The map of fNIRS channels. (**A**) The distribution of PFC channels and brain regions according to the 3D digitizer. (**B**) Subjects performing the N-back test. (**C**) Channels layout in the PFC. (DLPFC, dorsolateral prefrontal cortex; FPC, frontopolar cortex; VLPFC, Ventrolateral prefrontal cortex; OFC, orbital frontal cortex).

**Figure 3 brainsci-13-00497-f003:**
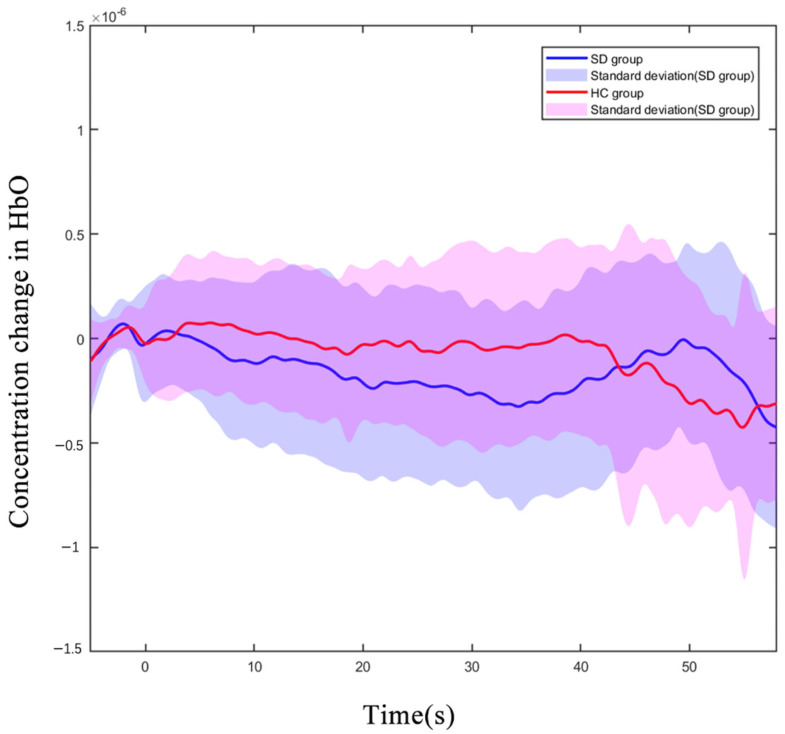
Oxygenated hemoglobin concentration changes in the HC and SD. Average oxy-Hb signal waveform of CH8 for 3 executions of 2-back task. The oxy-Hb signals of the SD group are displayed in blue, and the signals of the HC group are displayed in red. (X-axis: time process of executing 2-back tasks; Y-axis: oxygenated hemoglobin concentration level).

**Figure 4 brainsci-13-00497-f004:**
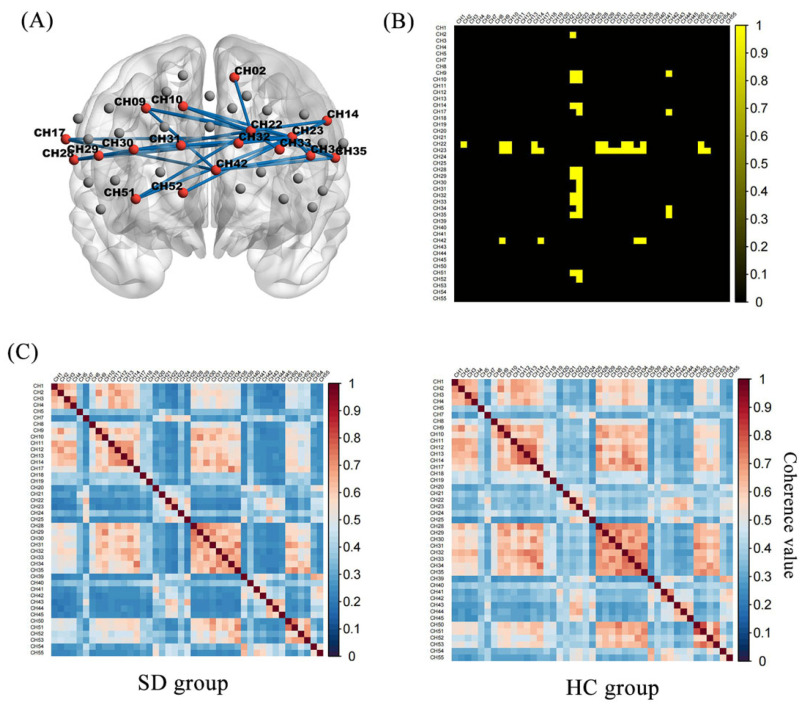
Functional connectivity matrix in the HC and SD: (**A**) Twenty-eight pairs had significantly higher channel-to-channel connectivity in the HC group than in the SD group. (**B**) The group-difference of channel-to-channel connectivity strength in binary matrices, where the yellow part is the significant difference part (**C**). Functional connection strength between the respective channel-to-channel of the SD and HC.

**Table 1 brainsci-13-00497-t001:** Data channel loss due to dark noise or saturation.

Subject	Excluded Channels	Excluded Channels Number/All
10	13	1/43
27	7, 17	2/43
31	2	1/43
56	7	1/43
58	1, 8	2/43

**Table 2 brainsci-13-00497-t002:** The demographics of the SD group and HC group.

	SD Group (*n* = 37)	HC Group (*n* = 37)	t/Z/χ^2^	*p*-Value
Age (years)	66.92 ± 3.59	66.86 ± 3.57	0.07	0.95
Education (year)	11.35 ± 1.96	11.41 ± 2.53	−0.10	0.92
Gender (male/female)	11/26	10/27	0.07	0.80
MoCA	27.00 (26.00–28.00)	27.00 (26.00–27.00)	−1.29	0.20
GDS	5.00 (4.00–5.00)	4.00 (4.00–5.00)	−1.16	0.25
Tea drinking situation				
Never	18	18	−0.112	0.911
Rarely	6	7		
Regular	13	12		
Coffee drinking situation				
Never	27	32	−1.427	0.154
Rarely	6	3		
Regular	4	2		

MoCA, Montreal Cognitive Assessment; GDS, Geriatric Depression Scale.

**Table 3 brainsci-13-00497-t003:** N-back tests performance of SD group and HC group.

	SD Group (*n* = 37)	HC Group (*n* = 37)	t/Z	*p*-Value	Effect Size
0-back Accuracy	97.62 (92.86–100.00)	100.00 (97.62–100.00)	2.15	0.03 *	0.26
0-back RT (ms)	571.49 (504.40–616.06)	567.60 (505.93–607.84)	0.06	0.95	−0.008
1-back Accuracy	90.48 (85.71–94.05)	90.48 (88.10–92.86)	0.43	0.67	−0.057
1-back RT (ms)	691.82 ± 110.67	685.06 ± 116.35	0.26	0.80	0.059
2-back Accuracy	83.59 ± 7.87	86.81 ± 6.14	−1.96	0.05 *	−0.456
2-back RT (ms)	798.29 (719.48–934.14)	833.90 (737.95–921.62)	0.33	0.74	−0.045

RT, Reaction Time; * *p ≤* 0.05.

## Data Availability

The data presented in this study are available on request from the corresponding author. The data are not publicly available due to our laboratory’s policy, we cannot provide raw data.

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
