# Peer review of "Prefrontal Cortex Hemodynamics and Functional Connectivity Changes during Performance Working Memory Tasks in Older Adults with Sleep Disorders"

_brainsci, 2023, doi:10.3390/brainsci13030497_

Round 1

Reviewer 1 Report

There needs some further explanations for the processes etc. For example in Figure 3, do the authors display average data, grand average, a case from the first stimulus of each subject and followed by a 30 s resting state? The explanation also needs to be aligned with figure 1.

The results do not provide sufficient information the deoxygenated hb and total blood values. Which are critical for fNIRS studies.

The reaction times for the harder test (2back) seems to be better performed by SD than HD which raises a question.

The explanation on the way the experiments conducted is very limited. Did the subjects all came and performed in the mornings? Did they refrain from tea, coffee etc? Did they have a good sleep the night before?

The overall study design is problematic as we do not know the current condition of the HC at the day of the experiments as well.

The behavioral data interpretation seems to be too strong given the fact that the results did not provide a clear explanation why the SD might have a better or worse performance of RT or scores differently for 0, 1, and 2 (N back) cases.

  It is hard to appreciate figure 4 as it is not clear what it indicates.  

Author Response

Point 1: There needs some further explanations for the processes etc. For example in Figure 3, do the authors display average data, grand average, a case from the first stimulus of each subject and followed by a 30 s resting state? The explanation also needs to be aligned with figure 1.

Response 1: Thank you for pointing this out, we have revised the legend for Figure 3. Figure 3 illustrates the time course of the average data for the three 2-back tasks for both groups of subjects in ch8, which contains a 10-s supply oxygen fallback for the resting state.

Point 2: The results do not provide sufficient information the deoxygenated hb and total blood values. Which are critical for fNIRS studies.

Response 2: Thank you for the suggestion. In response to your question, we have analysed HbR and HbT, and presented the results in the supplementary material. HbT showed significant group differences in channel 8 during the execution of 2-back. However HbR in channel 8 during 2-back did not show significant differences. We believe this is due to among the three NIRS parameters (HbO\HbR\HbT) measured, the concentration of HbO was found to be the most sensitive to changes in regional cerebral blood flow, which provided the strongest correlation with the blood oxygen level-dependent signal [1]. Therefore, we show only the oxyhemoglobin concentration in the manuscript and explain it in the manuscript for this practice(page 6,line192-196).

[1] Moriguchi Y, Hiraki K. Neural origin of cognitive shifting in young children. Proc Natl Acad Sci U S A. 2009 Apr 7;106(14):6017-21. doi: 10.1073/pnas.0809747106. Epub 2009 Mar 30.

Point 3: The reaction times for the harder test (2back) seems to be better performed by SD than HD which raises a question.

Response 3: Thank you for your question. In a review of previous studies, some researchers have suggested that subjects make a "trade off" between speed and correctness rate. Accordingly, we suggest that the more difficult 2-back required more cognitive ability, which may have led the HC group to try to maintain correctness and "trade off" between response time and correctness, prolonging response time. (page 10,line289-293)

[1] Zhang H, Zhang S, Lu J, et al. Social exclusion increases the executive function of attention networks [J]. Sci Rep, 2021, 11(1): 9494.

Point 4: The explanation on the way the experiments conducted is very limited. Did the subjects all came and performed in the mornings? Did they refrain from tea, coffee etc? Did they have a good sleep the night before?

Response 4: Thank you for your question.

  1. For the question " Did the subjects all came and performed in the mornings?",to ensure that subjects were performing at the same time, our FNIRS data collection was all performed in the morning.
  2. For the question " Did they refrain from tea, coffee etc? ",we had the daily habits of the subjects collected and statistically analyzed during the study. The results showed no between-group differences in tea and coffee consumption between the two groups of subjects, and the results have been presented in Table 2 for additional information.

SD group

N=37

HC group

N=37

Z

P‐value

Tea drinking situation

Never

18

18

-0.112

0.911

Rarely

6

7

Regular

13

12

Coffee drinking situation

Never

27

32

-1.427

0.154

Rarely

6

3

Regular

4

2

3.For the question " Did they have a good sleep the night before?”,we are not concerned with immediate effects, but rather with exploring the changes in brain function in the natural state of the two groups of subjects, in a relatively stable state over time, so we do not correct for the subjects' daily habits. Our PSQI scale already provides an objective assessment of the subjects' daily sleep status in the last month.

Point 5: The overall study design is problematic as we do not know the current condition of the HC at the day of the experiments as well.

Response 5 Thank you for pointing this out, The HC group were all elderly people with PSQI scale scores ≤5, which is an objective indicator scale that assesses the sleep condition of the subject in the past month and reflects the subject's recent sleep status, so we believe that the condition of the HC group on the day of the experiment should be relatively stable.

Point 6: The behavioral data interpretation seems to be too strong given the fact that the results did not provide a clear explanation why the SD might have a better or worse performance of RT or scores differently for 0, 1, and 2 (N back) cases.

Response 6: The issues you raised are very important to us and we have added to the discussion(page10,line286-294).

In our behavioral data, although there were no significant group differences in 0-back and 1-back response times, there was a trend toward poorer response times in the SD group. The higher working memory load of 2-back response time showed a better trend in the SD group, which, combined with the 2-back correct performance, we believe is a "trade off" behavior of the HC group, spending more time for a higher correct rate.

For the case of "scores differently for 0, 1, and 2 (N back) cases", we believe that as the number of items stored in working memory increases, the working memory load increases and performance is more susceptible to interference. Therefore, the scores for 0-back, 1-back and 2-back were different for the two groups of subjects, and both showed a trend of 0-back > 1-back > 2-back, which is consistent with the trend of the existing study results[1-2].

[1]  Fishburn F A, Norr M E, Medvedev A V, et al. Sensitivity of fNIRS to cognitive state and load [J]. Front Hum Neurosci, 2014, 8: 76.

[2]         Jacola L M, Willard V W, Ashford J M, et al. Clinical utility of the N-back task in functional neuroimaging studies of working memory [J]. J Clin Exp Neuropsychol, 2014, 36(8): 875-86.

Point 7: It is hard to appreciate figure 4 as it is not clear what it indicates.

Response 7: Thank you for underlining this deficiency. In response to your question, we have changed the legend to Figure 4.

Figure 4A shows the schematic diagram of the brain regions of the channel-to-channel connectivity with significantly different functional connectivity between the SD and HC groups.

Figure 4B shows a schematic diagram of the two-dimensional matrix of channel-to-channel connectivity with significant differences in functional connectivity between the SD and HC groups, where the yellow part is the significant difference part.

Figure 4C shows the functional connection strength between the respective channel-to-channel of the SD and HC groups.

Reviewer 2 Report

The paper reports about the modifications in the hemodynamics and functional connectivity in the prefrontal cortex during the administration of the n-back task in older adults with sleep disorders. The paper is interesting and well written, however some concerns should be addressed before publication:

1)     Please check some typos (e.g., line 51 and legend of Figure 3).

2)     Please check whether lines 176-177 are correct: “wavelet filtering, which was implemented in Homer2 hmrMotionCorrectSpline filtering function.”, because, maybe, this function provides a Spline based interpolation.

3)     In lines 204-206 it is stated that an independent sample t-test (or Mann-Whitney test for non-normal data) was performed on the hemodynamic concentration. Please, describe how the concentration was computed for statistical comparisons. In fact, from figure 3 the time courses of channel 8 for HC and SD seem quite overlapping. Moreover, please specify how the FDR was applied (how many comparisons have been considered?).

4)     How the Authors explain that no hemodynamic or connectivity differences were assessed during the 0-back task when the accuracies were significantly different between the two groups? It is true that the 0-back is more an attention task rather than a working memory task, but, maybe, differences in the prefrontal cortex should be assessed, since, as stated in line 39, attention is modulated by the prefrontal cortex.

Author Response

Point 1: Please check some typos (e.g., line 51 and legend of Figure 3).

Response 1: Thank you for your careful review, we have made changes in the text and highlighted them.

Line 51:the reduced functional connectivity(FC) of PFC subregions may underlie the deficits in working memory in patients with SDs.

legend of Figure 3: Oxygenated hemoglobin concentration changes in the HC and SD. Average oxy-Hb signal waveform of CH8 for 3 executions of 2-back task. The oxy-Hb signals of the SD group is displayed in blue, and the signals of the HC group is displayed in red. (X-axis: time process of executing 2-back tasks; Y-axis: oxygenated hemoglobin concentration level).

Point 2: Please check whether lines 176-177 are correct: “wavelet filtering, which was implemented in Homer2 hmrMotionCorrectSpline filtering function.”, because, maybe, this function provides a Spline based interpolation.

Response 2: The question you raised is very important for us. We have made changes in the manuscript and highlighted the changes.(page5,line175-176)

“Then the hmrMotionCorrectSpline function was used to create and subtract the splines as well as to add the vertical shift to realign the data.”

Point 3: In lines 204-206 it is stated that an independent sample t-test (or Mann-Whitney test for non-normal data) was performed on the hemodynamic concentration. Please, describe how the concentration was computed for statistical comparisons. In fact, from figure 3 the time courses of channel 8 for HC and SD seem quite overlapping. Moreover, please specify how the FDR was applied (how many comparisons have been considered?).

Response 3: Thank you for underlining this deficiency. We first obtained the mean hemoglobin concentration data for all individuals after averaging the time series in homer2, and then imported it into IBM SPSS Statistics version 24.0. The data were first tested for normality, and then the independent samples t-test was performed for those data that were normally distributed, and the Mann-Whitney test was performed for those data that were not normally distributed. The p-values obtained from the tests are presented in the supplementary material.

For the resulting P values FDR correction was performed in MATLAB, and data from all 43 channels of the prefrontal lobe were included in the FDR correction. The corrected data are also presented in the supplemental file.(page6,line203-214)

Point 4: How the Authors explain that no hemodynamic or connectivity differences were assessed during the 0-back task when the accuracies were significantly different between the two groups? It is true that the 0-back is more an attention task rather than a working memory task, but, maybe, differences in the prefrontal cortex should be assessed, since, as stated in line 39, attention is modulated by the prefrontal cortex.

Response 4: Thank you for pointing this out, we have added it to the discussion(page10,line321-331). A recent systematic review showed that patients with sleep disorders performed worse on attention tasks, which is the same situation as in our manuscript where there was a significant difference in 0-back correct rates [1]. Previous studies have shown that during the performance of a working memory task, participants performing a 1-back rather than a 0-back task had brain activity in working memory areas[2]. And another study also showed significantly longer feature path lengths in the 2-back task compared to the 0-back task [3]. Combining the above studies, we concluded that although attention is modulated by the prefrontal cortex, there was no significant difference in prefrontal activation between the two groups because the 2-back task was more challenging than the 0-back task, which required less cognitive demand for 0-back and less demand for prefrontal cortex activation.

[1]         Rodrigues T, Shigaeff N. Sleep disorders and attention: a systematic review [J]. Arq Neuropsiquiatr, 2022, 80(5): 530-8.

[2]         Konishi M, McLaren D G, Engen H, et al. Shaped by the Past: The Default Mode Network Supports Cognition that Is Independent of Immediate Perceptual Input [J]. PLoS One, 2015, 10(6): e0132209.

[3]         Sun J, Liu F, Wang H, et al. Connectivity properties in the prefrontal cortex during working memory: a near-infrared spectroscopy study [J]. J Biomed Opt, 2019, 24(5): 1-7.

Round 2

Reviewer 1 Report

The authors have addressed the questions adequately. There remain some points perhaps to be addressed by future research.

Reviewer 2 Report

I thank the Reviewer for addressing all my concerns. The manuscript is significantly improved, and, in my opinion, it is suitable for publication in the present form.